# IL-6, IL-12, and IL-23 STAT-Pathway Genetic Risk and Responsiveness of Lymphocytes in Patients with Multiple Sclerosis

**DOI:** 10.3390/cells8030285

**Published:** 2019-03-26

**Authors:** Marina R. von Essen, Helle B. Søndergaard, Eva R.S. Petersen, Finn Sellebjerg

**Affiliations:** The Danish Multiple Sclerosis Center, Department of Neurology, Rigshospitalet, University of Copenhagen, Blegdamsvej 9, 2100 Copenhagen, Denmark; Helle.Bach.Soendergaard@regionh.dk (H.B.S.); eva.rosa.sansone.petersen@regionh.dk (E.R.S.P.); finn.thorup.sellebjerg@regionh.dk (F.S.)

**Keywords:** STAT-pathway, genetic risk variants, multiple sclerosis

## Abstract

Multiple sclerosis (MS) is an immune-mediated demyelinating disease characterized by central nervous system (CNS) lymphocyte infiltration, abundant production of pro-inflammatory cytokines, and inappropriate activation of Th1 and Th17 cells, B cells, and innate immune cells. The etiology of MS is complex, and genetic factors contribute to disease susceptibility. Genome-wide association studies (GWAS) have revealed numerous MS-risk alleles in the IL-6/STAT3, IL-12/STAT4, and IL-23/STAT3-pathways implicated in the differentiation of Th1 and Th17 cells. In this study, we investigated the signaling properties of these pathways in T, B, and NK cells from patients with relapsing-remitting MS (RRMS) and healthy controls, and assessed the genetic contribution to the activity of the pathways. This revealed a great variability in the level of STAT-pathway molecules and STAT activation between the cell types investigated. We also found a strong donor variation in IL-6, IL-12, and IL-23 responsiveness of primed CD4+ T cells. This variation could not be explained by a single MS-risk variant in a pathway component, or by an accumulation of multiple STAT-pathway MS-risk SNPs. The data of this study suggests that other factors in cohesion with the genetic background contribute to the responsiveness of the IL-6/STAT3, IL-12/STAT4, and IL-23/STAT3-pathways.

## 1. Introduction

Multiple sclerosis is an immune-mediated, demyelinating disorder of the central nervous system (CNS). Immune dysregulation in the periphery of patients with multiple sclerosis (MS) is involved in the transmigration of lymphocytes to the CNS, resulting in myelin breakdown, axonal damage, and neuronal cell death [1]. Although incompletely understood, both environmental and genetic risk factors are thought to be involved in the peripheral immune dysregulation of MS patients [1]. Recent genome-wide association studies have revealed a number of MS risk alleles in or near genes encoding the Janus kinase (JAK)/signal transducers and activators of transcription (STAT) pathways activated by IL-12 and IL-23 signals. This includes the IL-12 and IL-23 shared cytokine receptor unit IL-12Rβ1, the JAK kinase TYK2, which is activated by both cytokines upon receptor binding, and STAT4 activated by IL-12 and STAT3 activated by IL-23 as well as the STAT3/4-pathway inhibitor SOCS1 [2,3,4]. In addition, MS risk alleles have been described for the cytokine subunits IL-12p35 and IL-12/23p40 themselves [2,3]. The IL-12/STAT4 pathway is critical for the differentiation of CD4+ T cells to Th1 cells, and the IL-23/STAT3 pathway is critical for the differentiation to Th17 cells. Th1 and Th17 cells are central to MS pathogenesis [1] and STAT3/STAT4 is essential for Th1/Th17-mediated CNS autoimmunity in animal models [5]. 

In addition to IL-23, IL-6 induces activation of STAT3. IL-6 signals through JAK1/JAK2/TYK2/STAT3 to induce expression of the IL-23 receptor and stabilizes the expression of RORγt, favoring Th17 differentiation [6]. Although both IL-6 and IL-23 employ STAT3 as their principal signaling moiety, and induce a Th17 phenotype of T cells, their downstream cellular effects are not identical. In addition to the MS-risk alleles shared with the IL-23/STAT3-pathway, the IL-6/STAT3-pathway includes MS-risk alleles near genes encoding the IL-6 receptor subunit IL6ST (gp130) and signaling molecule JAK1 [2,3,4]. 

With this study, we investigated the IL-12/STAT4, IL-23/STAT3, and IL-6/STAT3 pathways in patients with relapsing-remitting MS (RRMS) and the potential impact of genetic background on activation of these pathways. 

## 2. Materials and Methods

### 2.1. Study Population and Ethics

We included 18 healthy controls (mean 41 years; range 26–58) and 18 untreated patients with relapsing-remitting MS (RRMS) (mean 41 years; range 25–60). All patients were diagnosed with RRMS based on the 2010 revised McDonald criteria [7,8]. Untreated patients were defined as at least one month since last steroid treatment or more than three months since last immunomodulatory treatment. None of the patients had ever received strong immunosuppressive drugs, e.g., cyclophosphamide or mitoxantrone, or cell-depleting monoclonal antibody therapy. Healthy controls had no autoimmune, neurological, or other chronic illness. Samples from the patients with RRMS were collected on the same day as a sex-and-age matched control to minimize the variation between the two groups (mean age difference between collected pairs 3.6 years; no significant difference overall in age *p* = 0.92, *t*-test). All participants gave informed, written consent to participation. The study was approved by the regional scientific ethics committee (protocol number KF-01114309).

### 2.2. Blood Samples 

Venous blood was collected and peripheral blood mononuclear cells (PBMC) isolated by density gradient centrifugation with Lymphoprep (Axis-Shield, Oslo, Norway) and washed twice in cold PBS/2 mM EDTA. PBMC were either subjected to cytokine receptor staining by flow cytometry, STAT-pY analysis, or cultured for 6 days after which the STAT-pY analyses were repeated. PBMC were also cryopreserved until all samples were collected for mRNA analysis. Tissue collection and flow cytometric data acquisition from each patient with RRMS were performed in parallel with an age and sex-matched healthy control to minimize day-to-day variation between groups.

### 2.3. Cytokine Receptor Analysis 

PBMC were washed once in PBS and stained with live/dead stain (Life Technologies, Watham, MA, USA) according to manufacturer’s guidelines. Thereafter cells were washed and stained with fluorochrome-conjugated antibodies (Ab) against cytokine receptors or isotype matched controls: IL-6 Ra (PE; UV4; BioLegend, San Diego, CA, USA), IL-12Rb2 (APC; 305719; R&D Systems; Minneapolis, MN, USA), IL-23R (PE; 218213; R&D Systems). The cells were washed and finally stained with fluorochrome-conjugated Ab against lymphocyte markers: CD3 (FITC; SK7; BD Biosciences, Franklin Lakes, NJ, USA), CD4 (PE-Cy7; OKT4; BioLegend), CD8 (APC-AF750; 3B5; ThermoFischer Scientific, Watham, MA, USA), CD19 (APC-AF750; SJ25-C1; ThermoFischer Scientific), NKp46 (PE-Cy7; 9E2; BioLegend).

### 2.4. STAT-pY Analysis

Freshly isolated PBMC were either applied directly to STAT-pY analysis of T, B, and NK cells or primed for 6 days with human T activator beads (anti-CD3/CD28 beads; ThermoFischer Scientific) at a bead:cell ratio of 1:5 in serum free X-VIVO15 media (Lonza) prior to STAT-pY analysis of T cells. After 6 days of stimulation, the beads were removed, and the cells left to rest for 24 h. For the STAT-pY analysis, PBMC were stimulated for 20 min at 37 °C, 5% CO_2_ in serum free X-VIVO15 media supplemented with 100 ng/mL rhIL-6 (BD Biosciences), rhIL-12 (R&D Systems), rhIL-23 (R&D Systems), or medium alone. The cells were washed in ice cold PBS and stained with fluorochrome-conjugated Ab against: CD3 (FITC; SK7; BD Biosciences), CD4 (BV421; OKT4; BioLegend), CD8 (APC-AF750; 3B5; ThermoFischer Scientific), CD19 (APC-AF750; SJ25-C1; ThermoFischer Scientific) or NKp46 (BV421; 9E2; BioLegend). The cells were washed and incubated at 37 °C for 15 min at 37 °C in pre-warmed Lyse/Fix buffer (BD Biosciences). Thereafter cells were washed and incubated for 30 min on ice with −20 °C pre-cooled PermBuffer III (BD Biosciences). Cells were washed and incubated for 10 min at RT in normal mouse serum (DAKO) to reduce the background. STAT3-pY705 (AF647; 4/p-stat3; BD Biosciences) or STAT4-pY693 (PE; 38/p-stat4; BD Biosciences) Ab were added directly to the cells/mouse serum and further incubated for 50 min at RT. The cells were then washed and applied to flow cytometric analysis. 

### 2.5. Genotyping 

DNA was purified from 1 × 10^6^ pelleted and snap frozen PBMC using the Nucleospin tissue kit (Merchery-Nagel GmbH & Co. KG; Düren, Germany) according to the manufacturer’s guidelines. DNA purity and concentration were measured using the Nanodrop 2000 spectrophotometer (Thermo Scientific), and used for genotyping with commercially available TaqMan allelic discrimination assays with predesigned primers and probes on a ViiA7 instrument all from Thermo Fisher Scientific (Waltham, MA, USA). The following single nucleotide polymorphisms (SNPs) were genotyped in patients and healthy controls: rs740691 (*IL-12Rβ1*), rs34536443 (*TYK2*), rs1026916 (*STAT3*), rs6738544 (*STAT4*), rs12596260 (*SOCS1*), rs7731626 (*IL6ST*) and rs72922276 (*JAK1*). PCR and genotype scoring were performed as described by the manufacturer using TaqMan universal fast PCR master mix (Thermo Fisher Scientific). 

### 2.6. Fluorescence-Activated Cell Sorting of T, B, and NK Cells

Cryopreserved PBMC from patients and healthy controls were thawed and stained with fluorochrome-conjugated Ab against: CD3 (FITC; UCHT1), CD4 (BV421; OKT4), CD8 (APC; HIT8), CD19 (PE; HIB19), and NKp46 (PE-Cy7; 9E2) all from BioLegend. Hereafter, the PBMC were subjected to a 4-way fluorescence-activated cell sorting (FACS) into CD3+ CD4+ T cells, CD3+ CD8+ T cells, CD3- CD19+ B cells, and CD3- NKp46+ NK cells using a FACS Aria II (BD Biosciences, CA, USA). After FACS the cells were immediately lysed in Qiazol lysis buffer and frozen at −80 °C. The purity of isolated cells was 99.3% for CD4+ T cells, 99.5% for CD8+ T cells, 99.6% for B cells, and 98.1% for NK cells.

### 2.7. mRNA Analysis of T, B, and NK Cells 

RNA was purified from cryopreserved cell lysates from sorted CD4+ T cells, CD8+ T cells, B cells, and NK cells using the RNeasy Mini kit (Qiagen, Hilden, Germany, 74104) according to the manufacturer’s guidelines. RNA purity and concentration were measured using the Nanodrop 2000 spectrophotometer (Thermo Scientific) and 100 ng total RNA was used for measuring 770 mRNA targets on the nCounter PanCancer Immune Profiling Panel on a nCounter^®^ MAX Analysis System (NanoString Technologies, Seattle, WA, USA). Gene expression was normalized with 30 predefined reference genes included on the panel using the nSolver Analysis software. 

### 2.8. Pathway-Associated wGRS 

Pathway-associated weighted genetic risk scores (wGRS) were calculated based on the previously identified susceptibility variants in the IL-6/STAT3-pathway (IL6ST, JAK1, TYK2, SOCS1, STAT3), the IL-12/STAT4-pathway (IL12RB1, TYK2, SOCS1, STAT4), and the IL-23/STAT3-pathway (IL12RB1, TYK2, SOCS1, STAT3) (Appendix A). SNP odds ratios (ORs) were obtained from the combined analysis results from the MS replication study [4]. The natural logarithm to the ORs was used as weight for each risk allele. The number of risk alleles was counted for each individual and multiplied by the weight of the SNP. An individual wGRS was calculated as the sum of all weighted risk SNPs in the given pathway.

### 2.9. Statistical Analysis 

For comparison of HC and MS, a non-parametric Mann-Whitney *U* test was applied. For comparison of two genotypes, a Mann-Whitney *U* test was performed and for three genotypes a non-parametric Kruskal-Wallis and post-hoc Dunn’s test (uncorrected) was performed. Correlations were assessed by Spearman rank correlation analysis. *p* < 0.005 was considered significant, and *p* < 0.05 as suggestive of significance. 

## 3. Results

### 3.1. Association Between MS-Risk Alleles and Expression Level of Molecules in the IL-6, IL-12, and IL-23 Induced STAT-Pathway 

Genome-wide association studies (GWAS) have shown a striking coincidence of MS-risk alleles in the IL-6-, IL-12-, and IL-23-induced STAT-pathways in patients with RRMS [2,3,4]. To investigate if the expression level of STAT-pathway signaling molecules were associated with the genetic variant of the gene in question, we purified T, B, and NK cells from 36 genotyped individuals (healthy controls and patients with RRMS) and measured the expression level of JAK1, TYK2, STAT3, STAT4, and SOCS1. For cell separation, the surface markers CD3 (plus CD4 or CD8), CD19, and NKp46 were used for identification of T, B, and NK cells, respectively (Figure 1A–E). NKp46 was used in place of CD56 for NK cell identification, as CD56 could not be detected following the fixation process used for the STAT activity measurement described later in the paper. More than 80% of NK cells expressed NKp46, including the population of cytotoxic CD16+CD56dim cells and the CD56hi cells (data not shown). Analyzing the expression levels of JAK1, TYK2, STAT4, STAT5, and SOCS1 in these lymphocytes showed an association between the JAK1 risk-SNP rs72922276 and an increased level of JAK1-mRNA in CD8+ T cells (*p* < 0.0001; Figure 1F); the TYK2 risk-SNP rs34536443 and a decreased level of TYK2-mRNA in CD4+ T cells (*p* < 0.002; Figure 1G); and the *ST*AT4 MS-risk SNP rs6738544 and an increased level of STAT4-mRNA in CD8+ T cells (*p* < 0.004; Figure 1I). Except a suggestive association between the STAT3 MS-risk SNP rs1026916 and an increased level of STAT3-mRNA in B cells (*p* = 0.040; Figure 1H), no association was found in B or NK cells (Figure 1F–J). These observations suggest a MS-risk SNP-associated modulation of the IL-6-, IL-12-, and IL-23-induced STAT-pathways.

### 3.2. No Difference in the Expression of the IL-6R, IL-12R, and IL-23R in T, B, and NK Cells Between Patients with MS and Healthy Controls

To investigate a possible difference in the sensitivity of the IL-6-, IL-12-, and IL-23-induced STAT-pathways between patients with RRMS and healthy controls, we analyzed the expression of the IL-6 receptor (IL-6R), IL-12 receptor (IL-12R), and IL-23 receptor (IL-23R). Interleukin receptors are multimeric complexes; the IL-6R is composed of IL6ST (gp130) and IL-6Rα subunits; the IL-12R of IL-12Rβ1 and IL-12Rβ2 subunits; and the IL-23R of IL-12Rβ1 and IL-23R subunits. Measuring the mRNA expression level of IL6ST, IL-6Rα, IL-12Rβ1, IL-12Rβ2, and IL-23R subunits showed no difference in either CD4+ T cells, CD8+ T cells, B cells, or NK cells between patients with RRMS and healthy controls (Figure 2A–E), nor was the expression of IL-12Rβ1associated with the IL-12Rβ1 MS-risk SNP rs740691 (data not shown). We next measured the cellular surface expression of IL-6R, IL-12R, and IL-23R, targeting the receptor specific subunits IL-6Rα, IL-12Rβ2, and IL-23R by flow cytometry. This analysis showed that the percentage of CD4+ T cells, CD8+ T cells, B cells, and NK cells expressing IL-6R, IL-12R, or IL-23R on the cell surface was comparable in patients with RRMS and healthy controls (Figure 2F–T), except for a significantly higher percentage of B cells from patients with RRMS expressing the IL-23R (*p* < 0.0048; Figure 2S). In addition, we found that the surface expression level (mean fluorescence intensity; MFI) of the IL-6R, IL-12R, or IL-23R was equivalent between patients and healthy controls in all lymphocytes (data not shown). 

### 3.3. Similar Expression Levels of STAT3/4-Pathway Molecules in T, B and NK Cells between Patients with MS and Healthy Controls 

The biological activity of IL-6, IL-12, and IL-23 binding to their respective receptors is propagated in the cell through JAK1, JAK2, TYK2, STAT3, STAT4 and controlled by SOCS1 [9]. We therefore analyzed the expression level of these molecules in patients with RRMS and healthy controls in both T, B, and NK cells. This showed, that STAT3 was highly expressed by CD4+ T cells (Figure 3E), and JAK1 and STAT4 by NK cells (Figure 3A,F). In contrast, B cells only expressed low levels of STAT4, but had a high expression level of JAK2 and SOCS1 (Figure 3B,D,F). Comparing the level of JAK1, JAK2, TYK2, STAT3, STAT4, and SOCS1 between patients with RRMS and healthy controls, however, showed no statistically significant differences for T, B, or NK cells (Figure 3A–F). 

### 3.4. STAT Activation Induced by IL-6, IL-12, and IL-23 in Resting T, B and NK Cells from Patients with MS and Healthy Controls

To investigate the activity of the STAT3 and STAT4-pathways in T, B, and NK cells from patients with RRMS and healthy controls, we stimulated freshly isolated PBMC in culture for 20 min with either 100 ng/mL IL-6, IL-12, IL-23, or medium alone. The induced STAT-pathway activation was measured as phosphorylation of STAT3 and STAT4 molecules (pY). The mean frequency of STAT-pY+ unstimulated cells was 0.2%. The levels of STAT-pY in the cytokine stimulated cells were defined according to an isotype control and a gate set to include 0.2% of STAT-pY+ cells in the unstimulated sample. Except for a tendency of an increased STAT3 activation (STAT3-pY) in response to IL-23 in NK cells from patients with RRMS compared to healthy controls (*p* = 0.0108; Figure 4L), no difference was found in either IL-6-, IL-12-, or IL-23-induced activation of STAT3 and STAT4 in lymphocytes from patients with RRMS compared to healthy controls (Figure 4A–L). Despite IL-6R expression on NK cells (Figure 2), no IL-6-induced STAT3-pY was observed in these cells (data not shown). Likewise, B cells expressed both IL-12R and IL-23R (Figure 2), but no IL-12/IL-23-induced STAT4/STAT3-pY was observed in B cells (data not shown).

### 3.5. STAT Activation Induced by IL-6, IL-12, and IL-23 in Primed T Cells from Patients with MS and Healthy Controls

The sensitivity of the JAK/STAT-pathways are likely to change following priming of lymphocytes. We therefore primed T cells from patients with RRMS and healthy controls with a polyclonal stimulus (anti-CD3/CD28 beads) for 6 days and thereafter re-stimulated the cells for 20 min with either 100 ng/mL IL-6, IL-12, IL-23, or medium alone to induce phosphorylation of STAT3 and STAT4. The mean frequency of STAT-pY+ cells after 6 days in culture was 1%. The level of STAT-pY in the cytokine-stimulated cells was defined according to an isotype control and a gate set to include 1% of STAT-pY+ cells in the 6-day primed cells. This analysis showed that priming of CD4+ T cells increased the activation potential of STAT4 in response to IL-12 10-fold, and the activation of STAT3 in response to IL-23 5-fold. Priming of CD8+ T cells increased the activation potential of STAT4 in response to IL-12 5-fold. In contrast, no difference in STAT activation of primed versus resting T cells was found in response to IL-6. Comparing the activation of STAT3 and STAT4 in primed CD4+ and CD8+ T cells between patients with RRMS and healthy controls showed no difference (Figure 5A–I).

### 3.6. STAT3/STAT4 MS-Risk Alleles Are Not Associated with the Level of STAT3-pY/STAT4-pY

In patients with lupus erythematosus (SLE), a STAT4 SLE disease risk variant has been associated with increased sensitivity of IL-12-induced activity of the STAT4 signaling pathway in primed T cells [10]. We therefore hypothesized that the STAT4 MS-risk SNP rs6738544 was associated with the level of IL-12-induced STAT4-pY, and the STAT3 MS-risk SNP rs1026916 with the level of IL-6- and IL-23-induced STAT3-pY. The STAT3 and STAT4 genotypes of patients with RRMS and healthy controls included in our study were therefore correlated to IL-12-induced STAT4-pY and to IL-6- and IL-23-induced STAT3-pY. Neither in resting CD4+ T cells, CD8+ T cells, B cells, and NK cells, nor in primed CD4+ T cells and CD8+ T cells, did we find any association between STAT3-pY and STAT4-pY and the respective risk alleles (Figure 6A–O). 

### 3.7. Association between IL-6, IL-12, and IL-23 Responsiveness of Primed CD4+ and CD8+ T cells

We next investigated a possible association between the IL-6-, IL-12-, and IL-23-induced STAT3 and STAT4 activation in CD4+ and CD8+ T cells. This showed a strong positive correlation between the IL-12 and IL-23 sensitivity of primed CD4+ T cells (from patients and healthy controls) measured as STAT4-pY and STAT3-pY (*p* < 0.0001; Figure 7A). In contrast, we found a negative correlation between the sensitivity of IL-12 and IL-6 (*p* = 0.0009; Figure 7B) and between IL-23 and IL-6 (*p* = 0.0095; Figure 7C) in primed CD4+ T cells. In primed CD8+ T cells, we found a positive correlation between the sensitivity to IL-12 and IL-6 (*p* = 0.0028; Figure 7E). This observation indicates a common denominator for T cells from the same donor in relation to STAT-pathways. 

### 3.8. IL-6, IL-12, and IL-23 STAT-Pathway wGRS and Responsiveness of Primed T Cells

To investigate whether a coincidence of multiple MS-risk alleles in the IL-6/STAT3 pathway, IL-12/STAT4, and IL-23/STAT3 correlate with the sensitivity of the individual pathways, we calculated a pathway-specific weighted genetic risk score (wGRS). The MS-risk alleles associated with IL6ST (gp130), JAK1, TYK2, STAT3, and SOCS1 were included in the IL-6/STAT3-pathway wGRS; IL12RB1, TYK2, STAT4, SOCS1 in the IL-12/STAT4-pathway wGRS; and IL12RB1, TYK2, STAT3, SOCS1 in the IL-23/STAT3-pathway wGRS (Appendix A). Except for a weak correlation between IL-23 sensitivity and the IL-23/STAT3-associated wGRS in primed CD8+ T cells (*p* = 0.046; Figure 7L), no association between any of the STAT-pathway-specific wGRSs and the cytokine sensitivity of the T cells was found (Figure 7G–L). 

## 4. Discussion

GWAS have shown a great coincidence of MS-risk alleles in the IL-6-, IL-12-, and IL-23-induced STAT3/4-pathways [2,3,4], suggesting a risk allele-induced change in the activity of these pathways. In coherence with this theory, we found an increased level of JAK1 and STAT4 in CD8+ T cells from donors homozygous for the JAK1 MS-risk SNP rs72922276 (GG) and the *STAT4* MS-risk SNP rs6738544 (CC), respectively. *STAT4* expression is associated with Th1/Tc1 cells [11] implicated in the pathogenesis of MS [1], and a MS-risk SNP-associated increase in STAT4 expression therefore might contribute to susceptibility to MS. We also observed a significantly lower level of TYK2 in CD4+ T cells from TYK2 risk-SNP rs34536443 (GG) carriers, compared to carriers of the protective minor allele C (CG). The major G allele of the TYK2 variant rs34536443 encodes a proline in the kinase domain of TYK2, whereas the minor allele C encodes an alanine. The amino acid change from a proline to an alanine reduces the activity of TYK2, lowering the potential for the cells to produce IFN-γ and IL-17, hence the minor C allele is considered MS disease-protective [12,13]. The higher expression of TYK2 in donors expressing the protective genotype (CG) therefore possibly is a compensatory mechanism to counteract the reduced kinase activity of TYK2 in these individuals. 

In accordance with a possible influence of MS-risk alleles on the activity of STAT-pathways, we investigated signaling properties of the IL-6- and IL-23-induced STAT3-pathway and of the IL-12-induced STAT4-pathway in T, B, and NK cells from patients with RRMS and healthy controls. Investigating the level of IL-6, IL-12, and IL-23 receptors to determine the cellular sensitivity to IL-6, IL-12, and IL-23, showed no difference in the mRNA-level or the surface expression level (MFI) of the IL-6R, IL-12R, or the IL-23R between patients with RRMS and healthy controls. Analyzing the frequency of lymphocytes expressing these cytokine receptors showed that an increased percentage of B cells from patients with RRMS expressed the IL-23R. The relevance of IL-23R expression on B cells is unknown and, surprisingly, our subsequent analysis of IL-23-induced activation of STAT3 (STAT3-pY) in resting B cells showed no activation of STAT3, suggesting that any role for IL-23/STAT3 signaling in B cells requires prior activation. Consistent with data from the literature, we also found that few resting T cells expressed the IL-23R, in contrast to the IL-6R, which was expressed by the majority of T cells. In coherence, we observed that IL-23 only induced a minor degree of STAT3 activation in resting T cells, whereas IL-6 induced a high level of STAT3 phosphorylation. A previous study reported that IL-6-induced STAT3 activation leads to upregulation of the IL-23R [14], presumably increasing the cells’ sensitivity to IL-23 upon activation. This observation is consistent with our finding of a markedly increased IL-23-induced phosphorylation of STAT3 in primed T cells. 

In contrast to the high level of IL-6-induced STAT3-pY observed in resting T cells, IL-6 did not induce STAT3 activation in NK cells, despite IL-6R expression on the cell surface and high levels of JAK1 and TYK2. It is possible that IL-6-induced STAT3 activation in resting NK cells is suppressed by one of various receptors or JAK-phosphatases [15], or that IL-6-induced activation of STAT1 attenuates IL-6-induced STAT3-pY [16]. It is also possible that NK cells serve as a reservoir of soluble IL-6R (sIL-6R). sIL-6R is generated by proteolytic cleavage of the membrane-bound IL-6R in response to inflammatory signals with the purpose of amplifying IL-6-mediated signaling in cells expressing IL6ST, for example enhancing T cell proliferation and trafficking [17]. In our study, we found that IL-6ST particularly is highly expressed in CD4+ T cells, making these cells sensitive to the increased levels of sIL-6R previously documented in the serum of patients with MS [18]. It would be interesting to address the hypothesis of NK cells as an sIL-6R reservoir as a mean to increase the activation of CD4+ T cells in MS in future studies.

Comparing the activation of STAT3 and STAT4 in resting T, B, and NK cells between patients with RRMS and healthy controls only showed a tendency of an increased IL-23-induced STAT3-pY in the NK cells of the patients. Likewise, the activation of STAT3 and STAT4 in primed CD4+ T cells and CD8+ T cells showed no difference between patients and controls. Intriguingly though, we found a very strong positive correlation between IL-12-induced STAT4-pY and IL-23-induced STAT3-pY (*p* < 0.0001), and a negative correlation between IL-12-induced STAT4-pY and IL-6-induced STAT3-pY (*p* = 0.0009), and between IL-23-induced STAT3-pY and IL-6-induced STAT3-pY (*p* = 0.0095) in primed CD4+ T cells. One possible contribution to this observed donor variation in IL-6/IL-12/IL-23 responsiveness of primed CD4+ T cells is the genetic background of the individuals. Investigating this possibility showed that there was no correlation between a single MS-risk variant of a pathway component and the pathway activity measured as STAT3-pY or STAT4-pY. We also considered if individuals with the greatest genetic burden, i.e. the highest accumulation of STAT-pathway MS-risk SNPs, were associated with a strong STAT activation. For this, we calculated a STAT-pathway specific wGRS for each of the IL-6/STAT3, IL-12/STAT4- and IL-23/STAT3-pathways. This analysis showed no significant correlation between the coincidence of MS-risk alleles in either of the IL-6/STAT3-, IL-12/STAT4-, or IL-23/STAT3-pathways and the activity of the pathway. This indicates that other factors than the genetic background contribute to the responsiveness and activity of STAT-pathways, for example, environmental factors, which have been shown to modulate both innate and adaptive immunity, as well as increasing the risk of developing MS [19]. Furthermore, in vivo, immune cells are under the influence of the local environment, in contrast to the setup of this study. The inflammatory conditions found in patients with RRMS are likely to increase the stimulation of lymphocytes compared to a healthy control, for example, the increased level of IL-6 [20] and IL-6ST [18] found in patients with RRMS likely enhances IL-6 signaling in CD4+ T cells. To overcome this obstacle in future studies, lymphocytes could be cultured in media, including autologous serum.

In this study, we have investigated the IL-6/STAT3-, IL-12/STAT4-, and IL-23/STAT3-pathways in MS and the contribution of the genetic background to the responsiveness of these pathways. Earlier studies have indicated JAK/STAT-pathways as a potential therapeutic target in MS [21,22], and our finding of a donor-specific increased activity of the IL-12- and IL-23-induced STAT-pathways strengthens the perception that certain individuals may benefit from treatment. Previously, a phase II study tested the efficacy of a neutralizing antibody against the p40 subunit of IL-12 and IL-23 (ustekinumab), with the intention to block IL-12 and IL-23 signaling pathways in patients with RRMS. After 23 weeks of treatment, they found no reduction in the cumulative number of gadolinium-enhancing T1-weighted lesions, their primary outcome [23]. Considering the donor-specific variation in STAT-pathway activity, it may be valuable to develop a method enabling the selection of possible treatment responders. With this study, we hoped to find an association between STAT-pathway specific wGRS and the activity of the STAT-pathway, as wGRS assessment could have been one method used to evaluate candidates for treatment. Unfortunately, this association was not observed, and therefore, future studies to investigate STAT-pathways are warranted.

## Figures and Tables

**Figure 1 cells-08-00285-f001:**
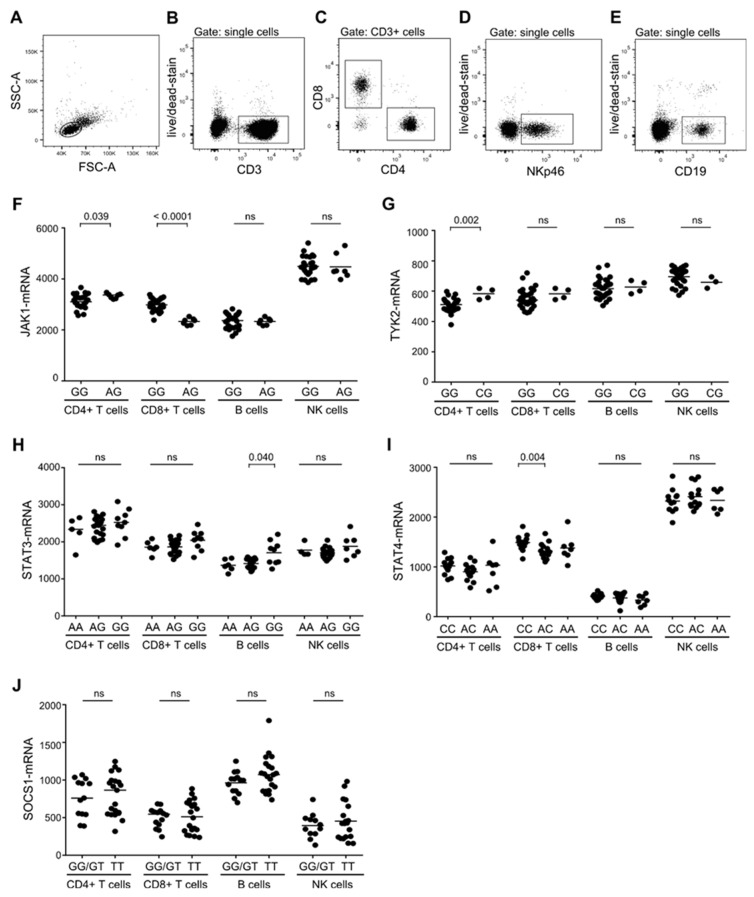
Multiple sclerosis (MS)-risk alleles and expression level of STAT-pathway molecules. (**A–E**) Gating strategy to identify T, B, and NK cells include a lymphocyte gate in a FSC-A/SSC-A dot plot (**A**), and a doublet cell exclusion in a FSC-A/FSC-H dot plot. T cells were then defined as CD3+ cells (**B**) and subdivided into CD4+ and CD8+ T cells (**C**). NK cells were defined as CD3- NKp46+ cells (**D**) and B cells as CD3- CD19+ cells (**E**). (**F**–**J**) mRNA level of JAK1 in donors homozygous (GG) or heterozygous (AG) for the JAK1 MS-risk allele rs729222 (**F**), of TYK2 in donors homozygous (GG) or heterozygous (CG) for the TYK2 MS-risk allele rs34536443 (**G**), of STAT3 in donors homozygous (AA), heterozygous (AG), or negative (GG) for the *STAT3* MS-risk allele rs1026916 (**H**), of STAT4 in donors homozygous (CC), heterozygous (AC) or negative (AA) for the STAT4 MS-risk allele rs6738544 (**I**), and of SOCS1 in donors homozygous/heterozygous (GG/GT) or negative (TT) for the SOCS1 MS-risk allele rs12596260 (**J**) in resting CD4+ T cells, CD8+ T cells, B cells and NK cells. The median value is shown for all groups analyzed.

**Figure 2 cells-08-00285-f002:**
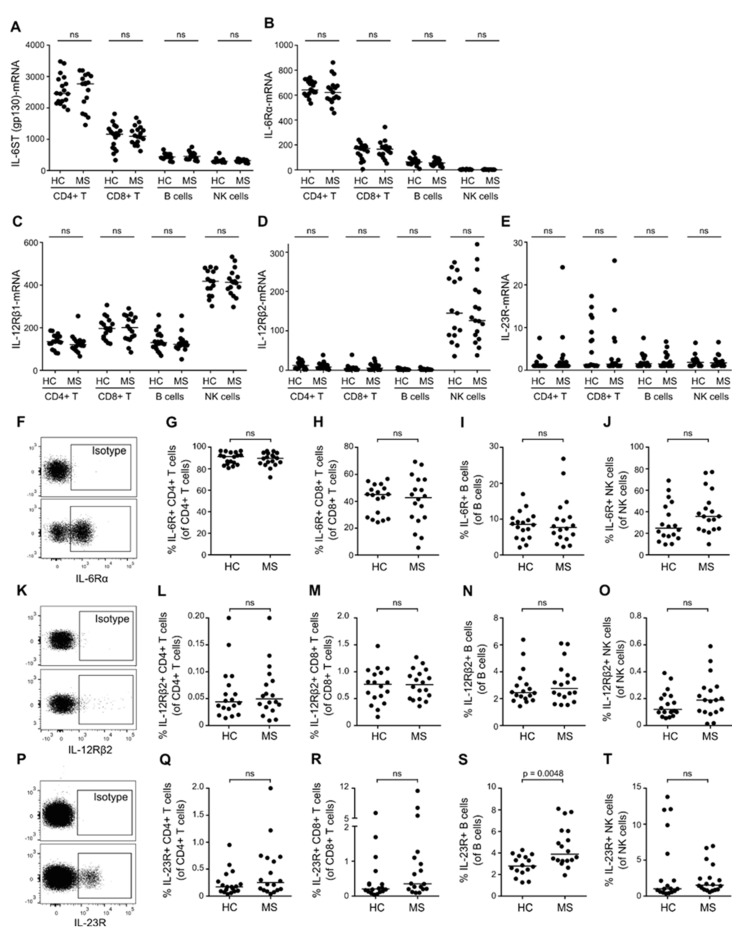
Expression of IL-6R, IL-12R, and IL-23R in T, B, and NK cells. (**A**–**E**) mRNA level of IL-6ST (**A**), IL-6Rα (**B**), IL-12Rβ1 (**C**), IL-12Rβ2 (**D**), and IL-23R (**E**) in CD4+ T cells, CD8+ T cells, B cells, and NK cells of healthy controls (HC) and patients with RRMS. (**F**) Dot plot example of IL-6Rα+ lymphocytes; isotype control is shown in the upper panel. (**G**–**J**) Frequency of IL-6Rα+ CD4+ T cells (**G**), CD8+ T cells (**H**), B cells (**I**), and NK cells (**J**) in healthy controls and patients with RRMS. (**K**) Dot plot example of IL-12Rβ2+ lymphocytes; isotype control is shown in the upper panel. (**L**–**O**) Frequency of IL-12Rβ2+ CD4+ T cells (**L**), CD8+ T cells (**M**), B cells (**N**), and NK cells (**O**) in healthy controls and patients with RRMS. (**P**) Dot plot example of IL-23R+ lymphocytes; isotype control is shown in the upper panel. (**Q**–**T**) Frequency of IL-23+ CD4+ T cells (**Q**), CD8+ T cells (**R**), B cells (**S**), and NK cells (**T**) in healthy controls and patients with RRMS. The median value is shown for all groups analyzed.

**Figure 3 cells-08-00285-f003:**
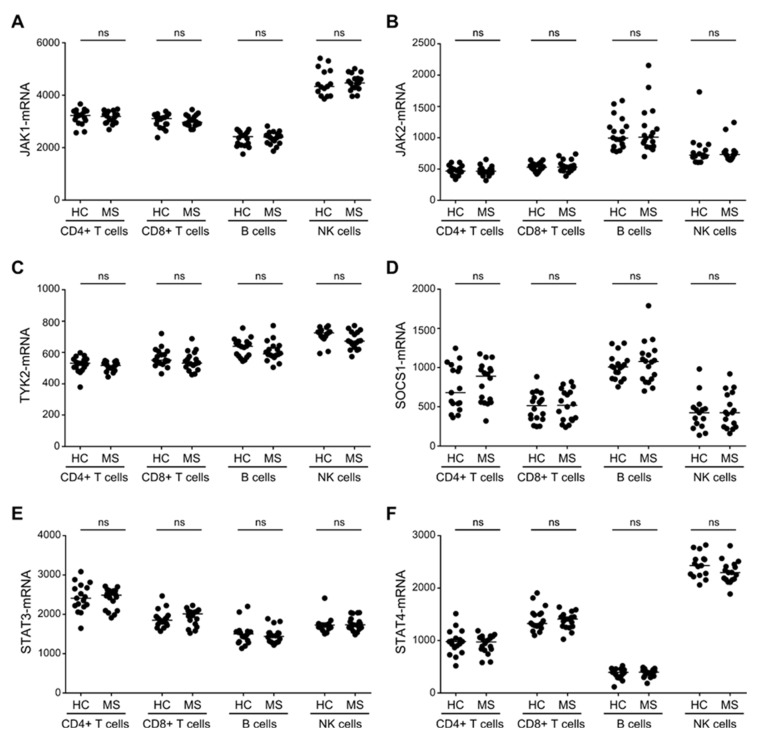
Expression levels of STAT3/4-pathway molecules in T, B, and NK cells. The mRNA level of JAK1 (**A**), JAK2 (**B**), TYK2 (**C**), SOCS1 (**D**), STAT3 (**E**), and STAT4 (**F**) in CD4+ T cells, CD8+ T cells, B cells, and NK cells of healthy controls (HC) and patients with RRMS. The median value is shown for all groups analyzed.

**Figure 4 cells-08-00285-f004:**
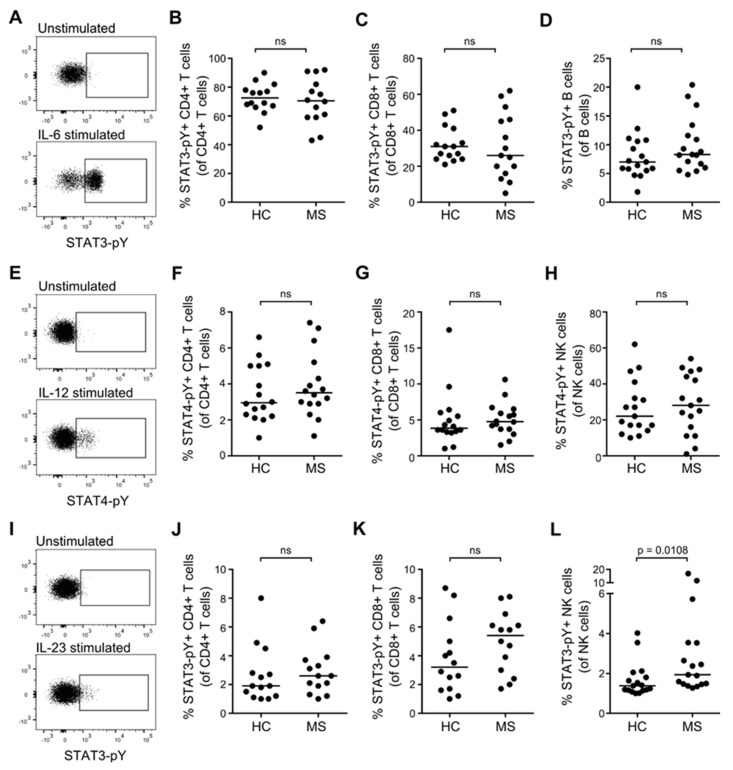
IL-6-, IL-12-, and IL-23-induced STAT activation in resting T, B, and NK cells. (**A**) Dot plot example of IL-6-induced STAT3-pY in resting lymphocytes; isotype control is shown in the upper panel. (**B**–**D**) Frequency of IL-6-induced STAT3-pY+ CD4+ T cells (**B**), CD8+ T cells (**C**), and B cells (**D**) in healthy controls (HC) and patients with RRMS. (**E**) Dot plot example of IL-12-induced STAT4-pY in resting lymphocytes; isotype control is shown in the upper panel. (**F**–**H**) Frequency of IL-12-induced STAT4-pY+ CD4+ T cells (**F**), CD8+ T cells (**G**), and NK cells (**H**) in healthy controls and patients with RRMS. (**I**) Dot plot example of IL-23-induced STAT3-pY+ in resting lymphocytes; isotype control is shown in the upper panel. (**J**–**L**) Frequency of IL-23-induced STAT3-pY+ CD4+ T cells (**J**), CD8+ T cells (**K**), and NK cells (**L**) in healthy controls and patients with RRMS. The median value is shown for all groups analyzed.

**Figure 5 cells-08-00285-f005:**
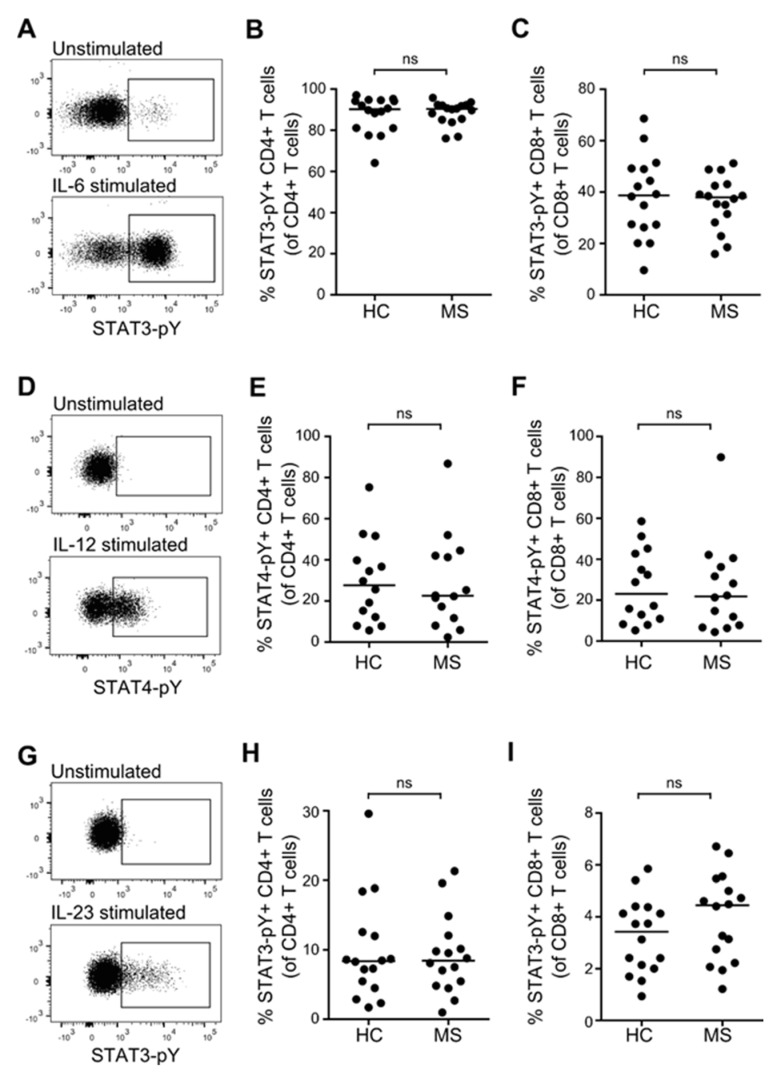
IL-6-, IL-12-, and IL-23-induced STAT activation in primed T cells. (**A**) Dot plot example of IL-6-induced STAT3-pY in primed T cells; isotype control is shown in the upper panel. (**B**,**C**) Frequency of IL-6-induced STAT3-pY+ CD4+ T cells (**B**) and CD8+ T cells (**C**) in healthy controls (HC) and patients with RRMS. (**D**) Dot plot example of IL-12-induced STAT4-pY in primed T cells; isotype control is shown in the upper panel. (**E**,**F**) Frequency of IL-12-induced STAT4-pY+ CD4+ T cells (**E**) and CD8+ T cells (**F**) in healthy controls and patients with RRMS. (**G**) Dot plot example of IL-23-induced STAT3-pY+ in primed T cells; isotype control is shown in the upper panel. (H-I) Frequency of IL-23-induced STAT3-pY+ CD4+ T cells (**H**) and CD8+ T cells (**I**) in healthy controls and patients with RRMS. The median value is shown for all groups analyzed.

**Figure 6 cells-08-00285-f006:**
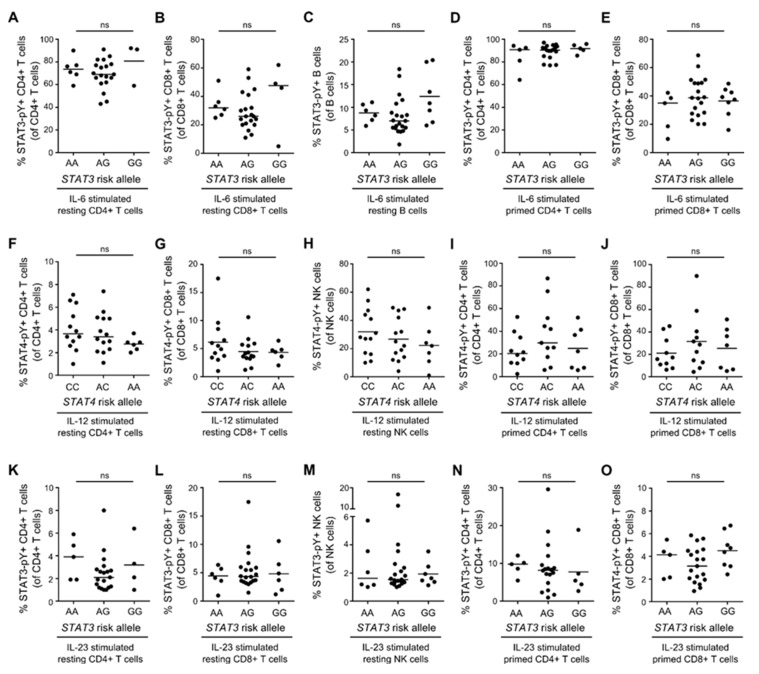
No association between STAT3/4 MS risk alleles and STAT3/4-pY. (**A**–**E**) Frequency of IL-6-induced STAT3-pY+ resting CD4+ T cells (**A**), resting CD8+ T cells (**B**), resting B cells (**C**), primed CD4+ T cells (**D**), and primed CD8+ T cells (**E**) in donors homozygous (AA), heterozygous (AG), or negative (GG) for the STAT3 MS risk allele rs1026916. (F-J) Frequency of IL-12-induced STAT4-pY+ resting CD4+ T cells (**F**), resting CD8+ T cells (**G**), resting NK cells (**H**), primed CD4+ T cells (**I**), and primed CD8+ T cells (**J**) in donors homozygous (CC), heterozygous (AC), or negative (AA) for the *STAT4* MS risk allele rs6738544. (**K**–**O**) Frequency of IL-23-induced STAT3-pY+ resting CD4+ T cells (**K**), resting CD8+ T cells (**L**), resting NK cells (**M**), primed CD4+ T cells (**N**), and primed CD8+ T cells (**O**) in donors homozygous (AA), heterozygous (AG), or negative (GG) for the STAT3 MS risk allele rs1026916. The median value is shown for all groups analyzed.

**Figure 7 cells-08-00285-f007:**
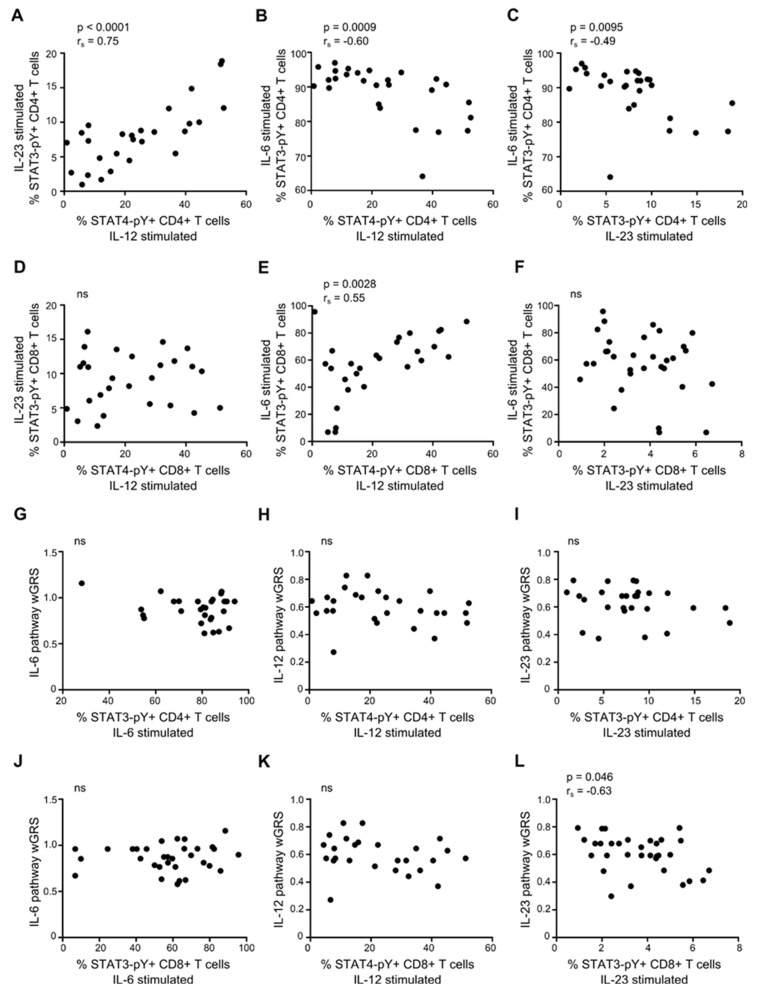
IL-6, IL-12, and IL-23 STAT-pathway weighted genetic risk score (wGRS) and responsiveness of primed T cells. (**A**–**C**) Correlation between IL-12-induced STAT4-pY and IL-23-induced STAT3-pY (**A**), between IL-12-induced STAT4-pY and IL-6-induced STAT3-pY (**B**), and between IL-23-induced STAT3-pY and IL-6-induced STAT3-pY (**C**) in primed CD4+ T cells. (**D**–**F**) Correlation between IL-12-induced STAT4-pY and IL-23-induced STAT3-pY (**D**), between IL-12-induced STAT4-pY and IL-6-induced STAT3-pY (**E**), and between IL-23-induced STAT3-pY and IL-6-induced STAT3-pY (**F**) in primed CD8+ T cells. (**G**–**I**) Correlation between the IL-6 pathway wGRS and IL-6-induced STAT3-pY (**G**), the IL-12 pathway wGRS and IL-12-induced STAT4-pY (**H**), and the IL-23 pathway wGRS and IL-23-induced STAT3-pY (**I**) in primed CD4+ T cells. (**J**–**L**) Correlation between the IL-6 pathway wGRS and IL-6-induced STAT3-pY (**J**), the IL-12 pathway wGRS and IL-12-induced STAT4-pY (**K**), and the IL-23 pathway wGRS and IL-23-induced STAT3-pY (**L**) in primed CD8+ T cells. The *p*-value and Spearman’s correlation coefficient r_s_ is shown for all plots.

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
