# Peer review of "IL-6, IL-12, and IL-23 STAT-Pathway Genetic Risk and Responsiveness of Lymphocytes in Patients with Multiple Sclerosis"

_cells, 2019, doi:10.3390/cells8030285_

Round 1

Reviewer 1 Report

Although the observations concerning genetic risk in IL-6, IL-12, and IL-23 in multiple sclerosis and strong donor variations in IL-6, IL-12, and IL-23 responsiveness are interesting, a number of points need clarifying and certain statements require further justification. These are given below.

1.    There are some reports concerning associations of interleukin gene polymorphisms (SNPs) and multiple sclerosis (MS). For example, Fedetz, M. et al.J. Neurol. Sci.190,69-72, 2001; Benešová, Y. et al.Acta Neurol. Belg.118, 493-501, 2018; Javan, M.R. et al.Neurol. Res.39, 435-441, 2017. The list should be much longer. The authors should include the papers and present a perspective. 

2.    There are some reports concerning associations of JAK/STAT pathway and MS. For example, Li, Y.et al.J. Immunol.192, 59-72, 2014; Benveniste, E.N. et al. J. Interferon Cytokine Res.34,577-588, 2014; Liu, Y. et al.Crit. Res. Immunol. 35,505-527, 2015. The authors should include the papers and present a perspective.

3.    The Instruction for Authors says, “Authors must use theMicrosoft Word templateorLaTeX templateto prepare their manuscript.”Judged by Figures and their legends, the authors did not use the temple. They should write the manuscript using the template.

1.    The authors should carefully read “Instruction for authors” and write accordingly. For example, in subsections such as 2.1., 2.2.•••etc., “Study population and ethics”should be “Study Population and Ethics”.

2.    Ref. 4, the author(s) should be “International Multiple Sclerosis Genetics Consortium.

3.    Ref. 5, the page number should be “e91663”.

Ref. 9, the authors’ names should be “Hammarén, H.M.; Virtanen, A.T.; Raivola, J.; Silvennoinen, O.”

Author Response

MAJOR POINTS

Point 1.

There are some reports concerning associations of interleukin gene polymorphisms (SNPs) and multiple sclerosis (MS). For example, Fedetz, M. et al.J. Neurol. Sci.190,69-72, 2001; Benešová, Y. et al.Acta Neurol. Belg.118, 493-501, 2018; Javan, M.R. et al.Neurol. Res.39, 435-441, 2017. The list should be much longer. The authors should include the papers and present a perspective. 

Comments from the author: The reviewer kindly suggests that more references on the subject of interleukin gene polymorphisms and MS are included. Two examples of investigations on IL-6 gene polymorphisms and MS performed in 131 and 297 patients, respectively, and one on IL-12B gene polymorphisms and MS performed in 145 patients are listed as examples. We appreciate the suggestion and acknowledge that there are many studies published on the subject which was not included. As references we have chosen larger studies involving >9000 cases collected in collaboration by more than 20 research groups, as genetic susceptibility studies are anticipated to require the analysis of sample sizes that are beyond the numbers currently available to individual research groups. Besides, the studies suggested by the reviewer finds that gene polymorphisms in the IL-6 and IL-12B gene is not associated with MS; which is in line with the larger studies used as references in this paper. For these reasons we did not include further references; hope the reviewer find this decision acceptable.

Point 2.

There are some reports concerning associations of JAK/STAT pathway and MS. For example, Li, Y.et al.J. Immunol.192, 59-72, 2014; Benveniste, E.N. et al. J. Interferon Cytokine Res.34,577-588, 2014; Liu, Y. et al.Crit. Res. Immunol. 35,505-527, 2015. The authors should include the papers and present a perspective.

Comments from the author: We thank the reviewer for this important comment and have accordingly included a perspective on the therapeutic value of JAK/STAT-pathway inhibition in patients with MS and included the references suggested by the reviewer in the Discussion section; line 364-377.

Point 3.

The Instruction for Authors says, “Authors must use theMicrosoft Word templateorLaTeX templateto prepare their manuscript.”Judged by Figures and their legends, the authors did not use the temple. They should write the manuscript using the template.

Comments from the author: It has been corrected in the manuscript.

MINOR POINTS

Point 1-3.

The authors should carefully read “Instruction for authors” and write accordingly. For example, in subsections such as 2.1., 2.2.•••etc., “Study population and ethics”should be “Study Population and Ethics”.

Ref. 4, the author(s) should be “International Multiple Sclerosis Genetics Consortium.

Ref. 5, the page number should be “e91663”.

Ref. 9, the authors’ names should be “Hammarén, H.M.; Virtanen, A.T.; Raivola, J.; Silvennoinen, O.”

Comments from the author: It has been corrected in the manuscript.

Reviewer 2 Report

The authors of “IL-6, IL-12 and IL-23 STAT-pathway genetic risk and responsiveness of lymphocytes in patients with multiple sclerosis” analyse the relevance of IL-12, IL-6, and IL-23 pathways in RR-MS patients, at different levels.

In particular, they analyse at DNA level the expression of two allele variants for JAK1, TYK2, STAT3, STAT4, and SOCS1 (genes involved in IL-12, IL-6, and IL-23 pathways), that were associated to MS risk.

Moreover, they compare the mRNA expression of those genes (JAK1, TYK2, STAT3, STAT4, SOCS1) and IL-12R, IL-6R, IL-23R in different immune cells, in MS versus healthy donors.

The activation of those pathways was evaluated in CD4, CD8, NK cells by analysis of the phosphorilation of STAT3, and STAT4 upon stimulation with cytokines.

In order to find an association between the genetic expression of STAT3 and STAT4 and their activation state, they calculated the expression of phospho STAT3 and phospho STAT4 for each allele variants. Then, they tried to find a correlation between the levels of STAT activation and the levels of STAT mRNA, or between the levels of STAT3 activation and STAT4 activation.  Finally, they calculated   the wGRS for the IL-6, IL-12, and IL-23 pathways and they correlated these levels with the levels of STAT phosphorilation.

1)Although most of the results reported in this manuscript are negative, this study is interesting and well done. However, a title more appropriated could be: “JAK1, TYK2, STAT3, STAT4 and SOCS1 genetic risk do not affect the responsiveness to IL-6, IL-12 and IL-23 in multiple sclerosis patients”

 2)It is not clear which cells are represented in the plots in Fig2 F, K, P. These panels should be replaced by representative plots for IL6R, IL23R, IL12R expression in CD4, CD8, B cells and NK cells, because these are the cells analysed in the cumulative data showed in G,H,I,J,L,M,N,O,Q,R,S,T.

3) Similarly to the previous issue, Figure 4 A,E,I should be replaced by plots representing  STAT3 and STAT4 phosphorilation upon IL-6, IL-12 and IL-23 stimulation in CD4, CD8, and NK cells.

4) Similarly to  previous issues, Figure 5A,D,G should be replaced by plots representing  STAT3 and STAT4 phosphorilation upon IL-6, IL-12 and IL-23 stimulation in CD4, and CD8 cells.

4)In order to verify the occurred STAT activation, the levels of phospho STAT3 and phospho STAT4 in unstimulated cells must be reported in the graph of Fig4 B,C,D,F,G,H,J,K,L, and Fig5 B,C, E,F,H,I.

5)In Figure 5 the levels of phospho STAT were evaluated on cells primed with antiCD3-28 for 6 days. It is not specified whether a period of starvation was applied to evaluate the subsequent STAT activation.

6)Figure 7 and 8 do not add any relevant data to the manuscript. I suggest deleting these figures, at least from the main figures.

Author Response

Point 1.

Although most of the results reported in this manuscript are negative, this study is interesting and well done. However, a title more appropriated could be: “JAK1, TYK2, STAT3, STAT4 and SOCS1 genetic risk do not affect the responsiveness to IL-6, IL-12 and IL-23 in multiple sclerosis patients”

Comments from the author: We thank the reviewer for his/her suggestion of a new title of the manuscript. We acknowledge the value of including the gene names of all the genes investigated in the title such as suggested; however, the title suggested has left out the genes IL6ST and IL12RB1. The title would therefore be very long and furthermore, does not take experiments performed without a correlation to the genetic risk into account. Based on this we find the current title more appropriate.

Point 2-4.

It is not clear which cells are represented in the plots in Fig2 F, K, P. These panels should be replaced by representative plots for IL6R, IL23R, IL12R expression in CD4, CD8, B cells and NK cells, because these are the cells analysed in the cumulative data showed in G,H,I,J,L,M,N,O,Q,R,S,T.

Similarly to the previous issue, Figure 4 A,E,I should be replaced by plots representing  STAT3 and STAT4 phosphorilation upon IL-6, IL-12 and IL-23 stimulation in CD4, CD8, and NK cells.

Similarly to previous issues, Figure 5A,D,G should be replaced by plots representing  STAT3 and STAT4 phosphorilation upon IL-6, IL-12 and IL-23 stimulation in CD4, and CD8 cells.

Comments from the author: The dot plots shown in figure 2, 4 and 5 are examples of flow cytometry staining of interleukin receptors or STAT-pY in the lymphocyte population (i.e. all lymphocytes) or T cells as described in the figure legend. The examples are used to illustrate gating strategies rather than specific staining of subpopulations of cells; the details of the individual cell types are given in the graphs. If an example of all cell types should be included in all figures, the figures would be almost double the current size, i.e. larger than one page; figure 1 would then include 12 flow cytometry dot plots in addition to the graphs shown, figure 4 would include 9 dot plots and figure 5 would include 6 dot plots. We did not include all of these 27 examples due to space limitations and as we did not believe they contributed with additional information. We hope the reviewer will find this decision satisfactory.

Point 4.

In order to verify the occurred STAT activation, the levels of phospho STAT3 and phospho STAT4 in unstimulated cells must be reported in the graph of Fig4 B,C,D,F,G,H,J,K,L, and Fig5 B,C, E,F,H,I.

Comments from the author: When measuring the level of phosphor STAT3 and STAT4 following stimulation with cytokines in figure 4 and 5, the STAT-phosphorylation levels were defined according to an isotype and the level of STAT-pY in unstimulated cells. The gate of STAT-pY+ cells after cytokine stimulation was set to include precisely 0.2 % and 1 % of STAT-pY+ cells in the unstimulated sample of freshly isolated and 6 days primed cells, respectively. We agree with the reviewer that this is an important point and therefore, the levels of phosphorylation in unstimulated cells and a more thorough description of the gating strategy has now been included in the result section 3.4 line 216-219 and section 3.5 line 246-248.

Point 5.

In Figure 5 the levels of phospho STAT were evaluated on cells primed with antiCD3-28 for 6 days. It is not specified whether a period of starvation was applied to evaluate the subsequent STAT activation.

Comments from the author: We thank the reviewer for this comment; the procedure has not been clearly stated in the manuscript. Following 6 days of stimulation with a polyclonal stimulus, the stimulus was removed, and the cells left to rest in serum free growth media X-VIVO15 for 24 h before STAT-pY analysis was performed. This information has now been included in the Materials and Methods section, line 91.

Point 6.

Figure 7 and 8 do not add any relevant data to the manuscript. I suggest deleting these figures, at least from the main figures.

Comments from the author: The reviewer suggests that figure 7 and 8 are deleted from the manuscript. We agree with the reviewer that data in figure 7 does not represent new or important data, and therefore, figure 7 and Result section 3.7 has been deleted from the paper. Figure 8 demonstrate that the level of STAT-pathway activity is donor specific. This observation led to the hypothesis that donor specific STAT-pathway activity corresponded to accumulation of STAT-pathway MS-risk alleles. By calculation of a donor specific STAT-pathway wGRS we challenged this hypothesis. If the hypothesis were true, calculation of STAT-pathway wGRS could potentially have been used in the selection of patients which would benefit from therapies based on JAK/STAT-pathway inhibition. A section in the Discussion section has been included to discuss the perspectives of JAK/STAT-pathway inhibition as a therapy for patients with MS in relation to our study; as suggested by Reviewer 1 (line 364-377).